# GNNs as Adapters for LLMs on Text-Attributed Graphs

## ABSTRACT

Text-attributed Graphs (TAGs), which interlace textual information with graph structures, pose unique challenges and opportunities for joint text and graph modeling. Recently, large language models (LLMs) have greatly advanced the generative and predictive power of text modeling. However, existing research on jointly modeling text and graph structures either incurs high computational costs or offers limited representational power. In this work, we propose *GraphAdapter* to harness the power of the LLM without fine-tuning its weights on Text-Attributed Graphs. Given a TAG, an adapter GNN is trained to reduce the LLM's error in predicting the next word of text sequences on nodes. Once trained, this GNN adapter can be seamlessly fine-tuned for various downstream tasks. Through extensive node classification experiments across multiple domains, *GraphAdapter* demonstrates an average improvement of ~5% while being more computationally efficient than baselines. We further validate its effectiveness with various language models, including RoBERTa, GPT-2, and Llama 2.

## KEYWORDS

Graph Neural Networks, Large Language Model, Text-Attributed Graph

## 1 INTRODUCTION

Graphs are ubiquitous in the real world [1]. In the past, graph structures have been extensively explored and utilized in many machine learning applications [27, 39]. In many practical cases, the nodes in graphs have textual features, known as Textual-Attributed Graphs (TAGs) [37]. For example, in social media [18], nodes represent users and node features are user profiles. Nodes in TAGs have both textual and structural data, which both reflect their intrinsic properties. Combining textual and structural data to modeling TAGs is an exciting new exploration for both graph machine learning and language modeling, which can benefit the application of graphs.

In TAGs, a complex correlation exists between the structural and textual data of nodes. Understanding this correlation can benefit the modeling of TAGs [5]. In Figure 1, user "Bob" frequently browses daily news on social media, as evidenced by the descriptions in his user profile. Users similar to Bob, who have many followers and often browse news nodes, are also likely interested in news. In other words, a graph can supplement textual attributes on a node through structural proximity. Graph Neural Networks (GNNs) are the de facto machine learning models for leveraging textual information alongside graph structures in TAGs. However, there's a lack of a unified GNN architecture compatible with different language models, especially the powerful foundation models.

Recently, there has been a surge in studies investigating effective ways to model both textual and structural data in TAGs. Some of these studies emphasize optimizing a cascading architecture that combines GNNs and LMs (**cascading GNN-LMs**) [37, 42]. One major challenge with these models is the extreme amount of additional computational cost brought by the message-passing mechanism.

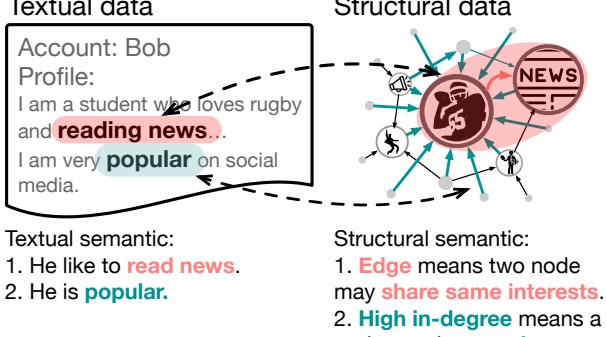

**Figure 1: An example of the correlation existing in the structural and textual data of nodes in social networks.**

To this end, several studies have successfully reduced the memory and computational overheads of such cascaded models by freezing partial or full parameters of the backbone language models [20, 25]. Large language models exhibit superior multi-task and few-shot learning capabilities across a wide spectrum of real-world applications [2]. However, when considering cascading GNN-LMs, existing techniques cannot be scaled up to billion-scale models like Llama 2 [33] and GPT-3 [2]. Another pioneering research has ventured to fine-tune language models using unsupervised graph information (**self-supervised GNN-LMs**) [4, 26]. For instance, GIANT [4] fine-tunes language models through a neighbor prediction task, subsequently using the refined language model to extract node representations for downstream tasks. These studies have conclusively shown that graphs can indeed aid language models in comprehending textual information. However, they separate the training of GNNs and LMs, potentially leading to sub-optimal graph-aware tuning results.

Instead of using graph information as supervision, we believe graph structure can enrich textual features through language modeling. In our previous example, structural proximity can be used to infer the user's preference even if he or she does not mention it in the profile. While cascading GNNs and LLMs prove infeasible for training, we draw inspiration from works on parameter-efficient tuning of LLMs to harness the power of large language models on TAGs [14, 22, 23] Therefore, we propose the use of GNNs as adapters for LLMs (i.e., *GraphAdapter*) offering several advantages:

- **Lightweight:** An GNN adapter introduces fewer than 1% of the trainable parameters compared to the popular Llama 2-7B model.
- **Convenience:** Given a pre-trained LLM and unlabeled graph, one can seamlessly integrate a graph-specific adapter for multiple graph applications.
- **Graph-aware tuning:** This enhances the predictive accuracy of the fine-tuned model by leveraging graph structures.

Now we present the details of *GraphAdapter* with respect to pre-training and fine-tuning of the adapter GNNs. To capture the data distribution of the graph, we employ parameter-efficient tuning of LLMs on node texts. This approach is similar to the continual training of language models [31] except GNN is the tuning parameter, which helps reduce the distribution discrepancy between the pre-training corpus and target data. To further improve the efficiency, we employ the GNN adapter only at the transformer's last layer and implement residual learning for autoregressive next token prediction. Different from a traditional adapter, we perform mean-pooling on the hidden representations from a GNN adapter and LLMs, then optimize the adapter to improve the next-word prediction of the LLMs. Once the adapter is trained, one can use *GraphAdapter* together with the backbone LLMs on various downstream tasks. For instance, we use a classification head atop the embeddings of the last token to fine-tune for node classification.

To verify the effectiveness of *GraphAdapter*, we conduct extensive experiments on multiple real-world TAGs including social and citation networks. *GraphAdapter* achieves an improvement of 4.7% over state-of-the-art cascaded GNN-LM methods and 5.4% over self-supervised GNN-LMs on average, with 30X fewer training parameters and storage. Moreover, once *GraphAdapter* is pre-trained, it can be conveniently fine-tuned for various tasks. Our ablation analysis shows that the pre-training step consistently improves the model performance across different graphs. We summarize our contributions as follows,

- *GraphAdapter* is a novel approach that harnesses the large language models on graph structure data with parameter-efficient tuning.
- We propose a residual learning procedure to pre-train the GNN adapter with the LLMs. The pre-training step significantly improves the fine-tuning performance of *GraphAdapter*.
- We conduct extensive experiments on large-scale TAGs using state-of-the-art open-sourced large language models (GPT-2 1.5B [28] and Llama 2 13B [33]). The results demonstrate that *GraphAdapter* can also reap the benefits of a larger model.

## 2 RELATED WORK

Modeling text-attributed graphs has attracted much attention in academia, which requires modeling both textual and structural data.

**Modeling semantics and graph structure**. Understanding the semantics is a key part of modeling TAG. Modeling semantics is a classic problem in natural language processing [43]. With the advent of Transformers [34], pre-trained language models have made breakthrough progress in modeling semantics [6]. These methods leverage massive unlabeled text through unsupervised methods like auto-regressive [19] and auto-encoding pre-training [12, 24] to train Transformers. The pre-training and fine-tuning paradigm emerged. However, fine-tuning has some limitations. Since language models have a large number of parameters, fine-tuning efficiency is low. There is also the problem of catastrophic forgetting. To solve these problems, some work proposed using adapter modules to reduce the number of parameters to fine-tune language models. For example, LoRA [14] trains a sparse matrix appended to the original parameters while keeping the language model frozen. Some work proposed using prompts to directly adapt language models

to downstream tasks without fine-tuning. However, prompts need manual design and do not provide stable improvements. Some work proposed prompt tuning [16, 23], which adds a trainable prompt and only trains the added prompt during training, greatly reducing the number of parameters. Another aspect of modeling language is modeling the structural information. With the proposal of GNNs [11], modeling graph structure achieved remarkable success. Many works [21, 36] have explored GNN architectures extensively, and these methods have achieved breakthrough progress in graph structure modeling.

**Modeling TAGs**. However, despite the success of language models and GNNs in their respective areas, how to utilize them to model text-attributed graphs still has many challenges. **(1) Cascading GNN-LM**: Directly cascading these two models is straightforward but has limitations, mainly high computational overhead. Since GNNs are mostly based on message-passing, they need to compute representations for many nodes simultaneously. Using language models to model so many text features requires huge memory and time costs. To address this, some work [25] proposed freezing the language model to reduce the computation needed for cascading. Some work [17, 20] proposed neighbor sampling but that reduces the graph information captured. Therefore, recently some work tried joint training of LMs and GNNs through knowledge distillation [26] or Expectation Maximization algorithms [42]. **(2) Self-supervised GNN-LMs**: some methods [4, 26] directly supervise language model fine-tuning through graph-related tasks, to help language models better understand the textual information in text-attributed graphs. The language model is then combined with GNNs by freezing the language model. This approach demonstrates the inherent connections between graph structure and text in TAGs. However, current research in this direction has limitations in that the LM and graph are separate, and the language model cannot directly perceive graph information. It also does not utilize the inherent connections between language and graphs to help GNNs better learn structural features. **(3) LLMs for Graph**: With the breakthrough progress made by LLMs on textual tasks [33, 41], recently many works have emerged exploring how to directly utilize LLMs to understand text-attributed graphs [3]. For example, by converting the graph to text [10, 40], or by converting it to a graph representation as part of a prompt [32]. Some works also explored using large models to enhance the textual features of text-attributed graphs [7, 13]. However, this paper is more focused on how to leverage the semantic information in text-attributed graphs to help us model text-attributed graphs. Therefore, this type of method not be further elaborated.

## 3 BACKGROUND

Before introducing the proposed method, it's important to understand some basic concepts and the background of pre-trained language models, graph neural networks, and text-attributed graphs.

### 3.1 Pretrained Language Model

**Textual data.** Textual data can be formulated as $\mathbb{D} = \{d_1, d_2...d_K\}$. It can be tokenized into a sequence of tokens $\mathbb{S} = \{s_1, s_2, ..., s_L\}$, where $s_i$ represents a specific token-id. In most cases, the first

token in the sentence (i.e., $s_0$) is [CLS], indicating the beginning of this sentence.

**Framework of PLMs**. A PLM consists of a multi-layer transformer encoder that takes a sentence $S_i$ as input and outputs the hidden states of each token:

$$\text{Transformer}(\{s_{i,0}, ..., s_{i,L}\}) = \{h_{i,0}, ..., h_{i,L}\}, \tag{1}$$

where $h_{i,k}$ is the dense hidden state of $s_{i,k}$.

**Pre-training of PLMs**. This paper uses the auto-regression task as an instance of pre-training, which is commonly applied to auto-regressive PLMs [29]. Given a sequence $\mathbb{S} = \{s_0, ..., s_L\}$, the goal is to model the joint probability of the sequence $P(\mathbb{S})$.

$$P(\mathbb{S}) = \prod_{k=1}^{L} p(s_i|s_0, ...s_{k-1}) \tag{2}$$

The transformer block is used to model these conditional probabilities. More specifically, at time step $k$ ($0 < k \leq L$), the transformer receives $\{s_0...s_{k-1}\}$ and outputs their hidden states $\{h_{i,0}, ..., h_{i,k}\}$. The $h_{i,k}$ are used to predict the probability distribution of the next token.

$$p(s_i|s_0, ...s_{k-1}) = \hat{s}_k = \sigma(\text{Head}(h_{i,k})) \tag{3}$$

The model parameters are trained to maximize the likelihood of $p(\mathbb{S})$, which is equivalent to minimizing the negative log-likelihood. Therefore, the loss function is:

$$\mathcal{L}_{LM} = \sum_{k=1}^{L} \text{CrossEntropy}(s_k, \hat{s}_k) \tag{4}$$

**Sentence representation**. Given a sentence $\mathbb{S}$ with length $L$, its sentence representation $W$ can be obtained by three methods [8, 30]: (1) first token representation, which uses the hidden state of the [CLS] token ($h_{i,0}$) as sentence representation. (2) mean-pooling representation, which is obtained by mean-pooling of all hidden states (i.e., $\text{Pool}(\{h_0...h_L\})$). (3) last token representation, which uses the hidden state of the last token.

**PLMs with prompts**. Due to the gap between pretraining tasks and downstream tasks, sentence representation may be hard to contain all the sentence information, thereby requiring fine-tuning for specific tasks. To address this issue, some studies utilize prompts to extract task-specific sentence features [16]. For example, suppose a $\mathbb{S}_i$ is a paper titled "Llama 2: Open Foundation and Fine-Tuned Chat Models", and the task is to classify the subject of it belongs. We can add some prompts to the sentence:

$$\{[Title], this, paper, belong, to, which, subject?\} \tag{5}$$

We denote this new sentence with the prompt inserted as $\mathbb{S}_{i|\mathbb{P}}$, where $\mathbb{P}$ represents the newly inserted tokens. We use the hidden state of the last token as the sentence representation, denoted as $W_{i|\mathbb{P}}$. Since the last token is used to predict the next token distribution in the pre-training stage, it can naturally combine the inserted prompt information into the original sentence and extract the prompt-related semantics. Extensive studies [19, 23] show that using prompts can reduce the gap between PLMs and downstream tasks and maximize the utilization of knowledge learned by PLMs during pre-training.

## 3.2 Graph Neural Network

Graph Neural Networks (GNNs) have achieved remarkable success in modeling graphs [9, 35]. The message-passing framework is a commonly used architecture of GNN.

**Graph.** Let $G = \{V, A\}$ denote a graph, where $V$ is the node set and $A$ is the adjacency matrix, with $A_{ij} = 1$ meaning there is an edge between node $i$ and node $j$. Usually, each node $i$ is associated with a node feature $x_i^0$.

**Framework of GNN.** The message-passing framework takes a set of node features $\mathcal{X} = \{x_i^0 | i \in V\}$, and an adjacency matrix $A$ as input and iteratively captures neighbors' information via pooling. More specifically, for a given node $i \in V$ in the $l$-th layer of message-passing, it can be formulated as:

$$x_i^l = f_2(\text{Pool}\{f_1(x_j^{l-1}|\theta_1^l)|j \in \mathcal{N}_i\}, x_i|\theta_2^l) \tag{6}$$

where $\text{Pool}\{\cdot\}$ is an aggregation function that combines the features of neighboring nodes, such as mean-pooling. And $\mathcal{N}_i$ denotes the set of neighbors of node $i$. Besides, $f_1(\cdot|\theta_1^l)$ and $f_2(\cdot|\theta_2^l)$ denote two trainable transformations with parameters $\theta_1^l$ and $\theta_2^l$ respectively. Further, we denote an $l_{max}$ layer message-passing framework as GNN, formally:

$$z_i = \text{GNN}(x_i^0, \mathcal{X}^0, A|\Theta_g) \tag{7}$$

where $z_i = x_i^{l_{max}}$, and $\Theta_g$ represents all the trainable parameters in the GNN. We use $z_i$ as the structural representation for node $i$.

## 3.3 Text-Attributed Graph

Let $\mathcal{G} = \{\mathcal{V}, \mathcal{A}\}$ denote a text-attributed graph, where $\mathcal{V}$ is the node set and $\mathcal{A}$ is the adjacency matrix. Each node $i \in \mathcal{V}$ is associated with a tokenized textual data, represented by $\mathbb{S}_i = \{s_{i,0}, ..., s_{i,L_i}\}$, which represents the textual data of the node.

**Problem Definition**: Give a text-attributed graph $\mathcal{G}$, the problem this paper focuses on is how to efficiently utilize the unlabeled textual data $\{\mathbb{S}_i | i \in \mathcal{V}\}$ in $\mathcal{G}$ to enhance the modeling of $\mathcal{G}$.

## 4 METHOD

This section introduces the proposed framework, referred to as *GraphAdapter*, which uses GNNs as adapters for LLMs to better model TAGs.

### 4.1 Overview

The core idea of GraphAdapter is: (1) combining GNNs as adapters to LMs. (2) pre-training GNN to align with LMs and enhance LMs through unlabeled textual data.

**Motivation:** In the textual data of TAGs, many structure-related semantics are hard to infer from context alone. As illustrated in the example in Figure 1, we can easily infer that this user is "popular" based on his degree in the social network, but it is difficult to infer from their description of habits alone. Combining structural information can enhance language models' ability to model these structure-related semantics in TAGs. Meanwhile, the process of enhancement is learning how to model structure. Therefore, the proposed method GraphAdapter, which first uses GNN as adapters for frozen PLMs, to combine structural information with PLMs, and

then pre-trains them through the semantic understanding task on TAGs.

**Language-structure pre-training**: In the field of natural language processing, pre-training is a common strategy used to self-supervised enhance language models' ability for semantic understanding, with techniques such as auto-regressive pre-training (e.g., GPT-2/3 [2, 29], Llama 2 [33], etc.) and auto-encoding pre-training (e.g., BERT[38], RoBERTa[24], etc.). Following our motivation, GraphAdapter uses the same pre-training task as these PLMs. To facilitate comprehension, this section only discusses GraphAdapter based on auto-regressive pre-training, and further details on how GraphAdapter is combined with other pre-training tasks can be found in the appendix. Since the pre-training process uses the context semantic to supervise structure learning, we refer to this pre-training as language-structure pre-training.

**Framework:** The framework of GraphAdapter is shown in Figure 2 (a). We also show how to fine-tune GraphAdapter on the downstream tasks in Figure 2 (b), we detail this part in Section 4.3. Given the textual data and graph structural data of a node, during the pre-training process, Step 1. GNN models the node structure information; Step 2. integrates the structural information with the corresponding context hidden-states modeled by PLM; and Step 3. predicts the next token. During this pre-training process, GraphAdapter can learn rich information. **Align GNN with the language model.** During the learning process, the node representation obtained by GNN is constantly combined with different representations modeled by the language model for reasoning, and the entire process naturally aligns these two. **Enhance GNN in modeling graph structure.** During the entire pre-training stage, the semantic information in the textual data supervises the GNN to model the graph structural information. **Better understanding the semantics in TAG**. GraphAapter can learn how to combine LLM and GNN to model the semantic information on TAG.

## 4.2 Pre-training on TAGs

In the training stage, GraphAdapter uses the textual data of each node in TAG to train GNN.

**Pipeline of pre-training**: Given a text-attributed graph $\mathcal{G}$, node $i$ and its textual data $\mathbb{S}_i = \{s_{i,0}, ..., s_{i,L_i}\}$, GraphAdapter uses all the tokens in $\mathbb{S}_i$ as supervision. For the $k$-th token, GraphAdapter first extracts its previous tokens $\mathcal{S}_{i,k} = \{s_{i,0}, ..., s_{i,k-1}\}$. Then, GNN models node $i$'s structure information $z_i$. The structure information is then combined with the previous tokens to predict the probability distribution of the next token, where the ground truth is token $s_{i,k}$.

**Structural representation**: GraphAdapter obtains its structural features $z_i$ through GNN. Here we use a general GNN based on the message-passing framework, which continuously aggregates neighbor features to obtain the new node's structural information. For whole process is formalized as:

$$z_i = \textbf{GNN}(x_i^0, \mathcal{X}, \mathcal{A}|\Theta_g) \tag{8}$$

where $x_i^0$ and $\mathcal{A}$ represent the initial node feature input and adjacency matrix in GNN, respectively. This paper used the sentence

representation of the corresponding node as $x_i^0$. See more details about GNN in Section 3.2.

**Context hidden-states**. GraphAdapter use the pre-trained transformer in PLM to encode $\mathcal{S}_{i,k}$, it is formalized as:

$$h_{i,k} = \textbf{Transformer}(\{s_{i,0}, s_{i,1}, ..., s_{i,k-1}\}) \tag{9}$$

Where the **Transformer**'s parameters are trained in frozen, and $h_{i,k}$ is the context hidden-states $\mathcal{S}_{i,k}$. Note that in the pretraining stage of PLM, $h_{i,k}$ is directly used to predict the next token, so $h_{i,k}$ contains both the context information and a certain of PLMs' prediction result.

**Fusion block**: GraphAdapter next fuse structural representation into context hidden-states, which is formalized as:

$$r_{i,k} = \textbf{Fusion}(h_{i,k}, z_i|\Theta_{fuse}), \tag{10}$$

The **Fusion**($*$) function is trainable with parameters $\Theta_{fuse}$. In this paper, MLPs are used as the structure of fusion. The process involves concatenating $h_{i,k}$ and $z_i$, and then feeding the resulting vector into MLPs.

**Residual connection**: the fused $r_{i,k}$ contains both structure information and context information. However, not every token's prediction requires the graph structure. For example, in the sentence "This paper focuses on graphs," the word "on" is simply a fixed collocation and easily inferred by context. Intuitively, words related to graph structure should be difficult for the language model to predict based on context. Therefore, the results of pre-trained language models are reused. We separately calculated the prediction probabilities of the language model alone and the probabilities that mixed the graph structure and the previous predictions. The two probabilities are then averaged to obtain the final prediction result. Formally:

$$\hat{s}_{i,k}^{LM} = \sigma(\textbf{Head}(h_{i,k})), \hat{s}_{i,k}^{GNN} = \sigma(\textbf{Head}(r_{i,k})) \tag{11}$$

$$\hat{s}_{i,k}^{ALL} = (\hat{s}_{i,k}^{LM} + \hat{s}_{i,k}^{GNN})/2 \tag{12}$$

Where $\sigma$ denotes the softmax function. Adding the original language model prediction results allows GNN to focus more on words that the language model cannot understand well.

**Optimization**: Our goal is to minimize the cross-entropy loss between the predicted probability distribution and the ground-truth distribution. Formally,

$$\mathcal{L}_{i,k} = \textbf{CrossEntropy}(\hat{s}_{i,k}^{ALL}, s_{i,k}) \tag{13}$$

$$\min_{\Theta_g, \Theta_{fuse}} \sum_{i \in V} \sum_{k \in \mathcal{S}_i} \mathcal{L}_{i,k} \tag{14}$$

Note, only **GNN**($*|\Theta_G$) and **Fusion**($*|\Theta_{fuse}$) of *GraphAdapter* are trainable in whole pre-training.

**GNN as Adapter**: In the whole pre-training stage, the GNN combines with the frozen LM's hidden states outputted from the transformer block. The combined hidden states are then input into the PLM's prediction head. Thus, the GNN acts as an adapter, altering the language model's predictions. Since the hidden states outputted

**Figure 2: Framework of *GraphAdapter*.** In the pre-training stage, Step 1. GNN models the node structure information, Step 2. integrates the structural information with the corresponding text fragment encoded by LM, and Step 3. predicts the masked token.

by the transformer block can be pre-processed and stored in advance. Therefore, the entire training process only requires training the GNN. Therefore, GraphAdpater can efficiently pre-train based on different scales of PLMs.

## 4.3 Fine-tuning with prompts

The pipeline is shown in Figure 2 (b). GraphAdapter is pre-trained by token-level semantic understanding tasks. To better utilize the learned knowledge of GraphAdapter and the PLMs in downstream tasks, we further proposed prompt-aware fine-tuning. It inserts prompts in textual data to get task-specific sentence embedding of each node. Prompts can transform various downstream tasks on TAGs into next token prediction. E.g., the task "*Which account is a student account*" can be transformed by a next-token prediction task, "[context], *based on this profile, this user is*". In the pre-training stage, GraphAdapter has learned how to utilize the structural information captured by GNN to enhance the accuracy of next-token prediction, therefore, under the transformed downstream task can better utilize the learned knowledge from pre-training. Formally, given textual data $\mathbb{S}_i$ of node $i$, we can combine a sequence of tokens with task-specific prompts behind textual data, namely, $\mathbb{S}_{i|\mathbb{P}} = [\mathbb{S}_i, \mathbb{P}]$, then we can get its sentence hidden states $h_{i|\mathbb{P}}$ through the transformer of PLM. The resulting hidden state is then fused with the node's structural representation as node representation in a specific downstream task.

$$r_{i|\mathbb{P}} = \text{Fusion}(h_{i|\mathbb{P}}, z_i) \qquad (15)$$

This node representation can be used in various tasks. For example, in the node classification, we can append a new linear transformation to output the result, i.e., $\hat{y}_{i|\mathbb{P}} = f(r_{i|\mathbb{P}}|\theta_{new})$. In fine-tuning stage, the whole parameters $\{\Theta_g, \Theta_{fuse}, \theta_{new}\}$ in GraphAdapter are trainable.

## 5 EXPERIMENT

To comprehensively validate that GraphAdapter can mine the intrinsic correlation between the textual and structure data in TAGs,

we conduct extensive experiments on three real-world datasets from diverse domains.

Our experimentation centered on the following five questions:

- *Q1*: How well is GraphAdapter in modeling TAGs?
- *Q2*: Whether GraphAdapter can adapt to other PLMs?
- *Q3*: Are all components comprising GraphAdapter valid?
- *Q4*: What exactly does GraphAdapter's pre-training learn?
- *Q5*: How efficient is GraphAdapter?

## 5.1 Experiment setup

**Dataset.** We select three public and real-world datasets used for evaluation: **Ogbn-arxiv[15]**: Ogbn-Arxiv (shorted as Arxiv), is a citation network where edges represent citation relationships, nodes represent papers and the text attribute is the abstracts of papers. The task on this graph is to predict paper subjects. **Instagram[18]**: Instagram is a social network where edges represent following relationships, nodes represent users, and the prediction task is to classify commercial and normal users in this network. The text attribute is the user's profile. **Reddit**[2]: Reddit is also a social network where each node denotes a user, the node features are the content of users' historically published subreddits, and edges denote whether two users have replied to each other. The prediction task is to classify whether a user is in the top 50% popular (average score of all subreddits). Table 1 shows detailed statistics of these datasets.

**Baselines.** We compare the proposed GraphAdapter with several state-of-the-art TAG modeling methods.

- **GNN-based methods**: This method directly combines different frozen PLM with GNNs to model TAGs. Since the specific GNN framework is not the key point this paper focuses on, this paper uses GraphSAGE [11] as an instance of GNN.
- **LM-based methods**: we select GIANT [4, 42], and GLEM as baseline. GIANT use self-supervised task to finetune PLM. Then incorporates the fine-tuned PLM and GNN to model TAG. GLEM

---

[2]https://convokit.cornell.edu/documentation/subreddit.html

**Table 1: Statistics of experiment datasets.**

| Dataset | # Nodes | # Edges | # Tokens | Split ratio (%) | #Class | Metric |
|---|---|---|---|---|---|---|
| Arxiv | 169,343 | 1,166,243 | 35,920,710 | 54/18/28 | 40 | Accuracy |
| Instagram | 11,339 | 144,010 | 579,263 | 10/10/80 | 2 | ROC-AUC |
| Reddit | 33,434 | 198,448 | 6,748,436 | 10/10/80 | 2 | Accuracy |

**Table 2: The performance of different methods across three datasets.** Each row corresponds to a specific method, and each column presents the performance of the models on a particular dataset. The evaluation metric used is accuracy for the Arxiv and Reddit datasets, and ROC-AUC for Instagram. The LM employed in each method is indicated in parentheses.

| | | Arxiv | Instagram | Reddit |
|---|---|---|---|---|
| LM | GNN (Ogb-feature) | 0.6980 (0.0013) | - | - |
| | GNN (RoBERTa) | 0.7129 (0.0013) | 0.6123 (0.0063) | 0.6191 (0.0043) |
| | GNN (RoBERTa+Prop) | 0.7067 (0.0011) | 0.6138 (0.0117) | 0.6198 (0.0036) |
| | GIANT (BERT) | 0.7262 (0.0011) | 0.5986 (0.0022) | 0.6379 (0.0045) |
| | GIANT (BERT+Prop) | 0.7252 (0.0012) | 0.6029 (0.0123) | 0.6348 (0.0039) |
| | GLEM[1] (DeBERTa) | 0.7550 (0.0024) | - | - |
| | GLEM (DeBERTa) | 0.7355 (0.0034) | 0.6166 (0.0056) | 0.6228 (0.0060) |
| | GLEM (DeBERTa+Prop) | 0.7315 (0.0033) | 0.6105 (0.0038) | 0.6221 (0.0052) |
| LLM | GNN (Llama 2) | 0.7305 (0.0020) | 0.6221 (0.0112) | 0.6320 (0.0041) |
| | GNN (Llama 2+Prop) | 0.7336 (0.0018) | 0.6312 (0.0051) | 0.6324 (0.0033) |
| | TAPE (GPT-3.5) | 0.7672 (0.0007) | - | - |
| Ours | GraphAdapter (w/o Pre) | 0.7648 (0.0020) | 0.6351 (0.0077) | 0.6284 (0.0025) |
| | **GraphAdapter** | **0.7707 (0.0015)** | **0.6513 (0.0075)** | **0.6461 (0.0019)** |

[1]performance reported in [42]

**Table 3: The performance of the GraphAdapter based on different LM across three datasets.** The evaluation metrics used for these datasets align with those outlined in Table 2.

| | Arxiv | | | Instagram | | | Reddit | | |
|---|---|---|---|---|---|---|---|---|---|
| | RoBERTa | GPT2 | Llama 2 | RoBERTa | GPT-2 | Llama 2 | RoBERTa | GPT-2 | Llama 2 |
| GNN (PLM) | 0.7129 (0.0013) | 0.7174 (0.0019) | 0.7305 (0.0022) | 0.6123 (0.0063) | 0.6019 (0.0124) | 0.6221 (0.0112) | 0.6191 (0.0043) | 0.6282 (0.0036) | 0.6320 (0.0041) |
| GNN (PLM+Prop) | 0.7067 (0.0011) | 0.6915 (0.0021) | 0.7336 (0.0027) | 0.6138 (0.0117) | 0.6128 (0.0014) | 0.6312 (0.0051) | 0.6198 (0.0036) | 0.6206 (0.0011) | 0.6324 (0.0033) |
| GraphAdapter (w/o Pre) | 0.7069 (0.0026) | 0.7146 (0.0025) | 0.7648 (0.0020) | 0.6165 (0.0038) | 0.6162 (0.0066) | 0.6351 (0.0077) | 0.6210 (0.0036) | 0.6284 (0.0027) | 0.6369 (0.0025) |
| GraphAdapter | **0.7273 (0.0021)** | **0.7325 (0.0022)** | **0.7707 (0.0015)** | **0.6292 (0.0033)** | **0.6276 (0.0034)** | **0.6508 (0.0033)** | **0.6379 (0.0061)** | **0.6441 (0.0022)** | **0.6461 (0.0019)** |

jointly trains PLM and GNN. Note, GIANT is based on BERT, and GLEM uses DeBERTa. Considering PLMs have a high influence on performance, we also compare GraphAdapter with them under the same PLM.

- **LLM-based methods**: There are a few LLM-based methods that are suitable in our setting. Therefore, we select TAPE [13] as the LLM-based baseline. This method, due to its need to obtain the interpretation of the text graph through GPT-3.5 and only the interpretation data on Arxiv is published. Therefore, we only report the results of this method on Arxiv.

Since many baseline methods involve GNN components, which are mostly optional, and considering that different GNNs have different performances. To make a fair comparison and without loss of generality, all GNNs used in all baselines are fixed to GraphSAGE, which is a classic and general GNN model.

**Prompts.** Since GraphAdapter involves prompts, to make a fair comparison, we also enhance the baselines with prompts. We provide detailed records of the prompts used in different experiments and how prompts are added to the baselines in the appendix. Meanwhile, we also validate the stability of our method with different prompts, and the experiment results can be found in the appendix. However, since prompts are not the main contribution of our method, this paper does not explore prompt methods such as soft-prompt and chain-of-thought in detail.

**Implementation details.** See more in the appendix.

## 5.2 Performance

### Q1: How well is GraphAdapter in modeling TAGs?

*A1*: **GraphAdapter can effectively model TAGs and surpass current state-of-the-art baselines on node classification tasks.** We compare GraphAdapter with 6 state-of-the-art baselines on 3 different real-world datasets to evaluate its effectiveness. As Table 2 shows, the experiment results suggest:

(1) Frozen LLMs are effective on TAGs. In general, frozen LLMs have an improved performance compared to the previous frozen LM. Experiment results show Llama 2 has improved performance on 3 datasets by 1.34% compared to RoBERTa-based methods. LLM can better combine the information in prompts to extract task-relevant sentence representations of nodes. As the results show, prompts can bring a 0.42% improvement on average for LLM, but they could not improve the performance of LM. Frozen LLMs with prompt can surpass many GNN-LM methods that require tuning LM. Results also show that LLMs with prompts can surpass GLEM and GIANT by 0.43% and 0.79% on average, respectively.

(2) Directly fusing GNN and LLM results in unstable improvements. Compared to ordinary GNN, GraphAdapter (w/o Pre) only adds one fusion component to fuse the semantic representation from the LM and structural representation from the GNN. Experiment results show that directly fusing language model representations only brings improvements on Arxiv, but not obviously on other datasets. Note that the Arxiv training samples are much larger than the other datasets. This result suggests that training samples may have an impact on GNNs to understand and effectively incorporate the representations inferred by LLMs with prompts.

(3) GraphAdapter can effectively combine GNN and LLM, surpassing existing state-of-the-art baselines in terms of performance. The pre-training effect of GraphAdapter is significant, bringing an average performance improvement of 1.98% and thus surpassing existing state-of-the-art baselines. Specifically, GraphAdapter achieves an improvement of 4.72% over state-of-the-art cascaded GNN-LM methods and 5.40% over self-supervised GNN-LMs on average. At the same time, GraphAdapter also surpasses TAPE, another LLM-based method on Arxiv by 0.4% accuracy improvement.

**Q2: Whether GraphAdapter can adapt to other PLMs?**

**A2: GraphAdapter can be effectively pre-trained based on RoB-ERTa, GPT-2, and Llama 2, resulting in performance improvements.** We run GraphAapter based on 3 different LM. The experiment results are shown in Table 3. GraphAdapter improved average performance over directly combining GNNs with frozen PLM by 1.67% on RoBERTa, 1.89% on GPT-2, and 2.77% on Llama 2. Meanwhile, GraphAdapter pre-training brings 1.67%, 1.50%, and 1.02% improvements on RoBERTa, GPT-2, and Llama 2 respectively. This result fully demonstrates that **GraphAdapter is a general and scalable method.** It is worth noting that the pre-training method of RoBERTa is different from others. GraphAdapter uses a pre-training task similar to RoBERTa, so there are some slight differences from the formula in Section 4. The main differences come from the loss function and language model inputs. We describe the details of applying GraphAdapter on Roberta in the appendix.

**Under the same PLM, the performance of GraphAdapter is comparable to the SOTA baselines based on fine-tuning the PLM.** We evaluate the performance difference between GraphAdapter and SOTA baselines under the same LM. Since the GLEM adopted DeBERTa, however, the pre-training code of DeBERTa is not open-sourced at present. To keep consistent, GraphAdapter and GLEM both adopt the same RoBERTa-base. As shown in Table 4, the experiment results suggest that methods based on pre-training like GIANT and GraphAdapter perform better on small datasets like

**Table 4: The performance of different methods using the same LMs across three datasets.** The evaluation metrics employed for these datasets align with those described in Table 2.

| | Arxiv | Instagram | Reddit |
|---|---|---|---|
| GNN (BERT) | 0.7039 (0.0013) | 0.5973 (0.0063) | 0.6061 (0.0043) |
| GIANT (BERT) | **0.7269** (0.0021) | 0.5986 (0.0022) | **0.6379** (0.0045) |
| GraphAdapter (BERT) | 0.7264 (0.0012) | **0.6156** (0.0032) | 0.6366 (0.0034) |
| GNN (RoBERTa) | 0.7129 (0.0013) | 0.6123 (0.0063) | 0.6191 (0.0043) |
| GLEM (RoBERTa) | **0.7308** (0.0029) | 0.6114 (0.0075) | 0.6228 (0.0018) |
| GraphAdapter (RoBERTa) | 0.7273 (0.0021) | **0.6276** (0.0034) | **0.6379** (0.0061) |

Instagram and Reddit. Similarly, Roberta-based GraphAdapter outperforms GLEM by 1.57% and BERT-based GIANT outperforms GLEM by 1.15% on small datasets. Compared to baselines based on pre-training, although GIANT fine-tunes the LM, its performance is 0.51% lower than GraphAdapter on average. Therefore, overall, even without fine-tuning the LM, the performance of GraphAdapter is comparable to current state-of-the-art baselines based on fine-tuning the LM.

## 5.3 In-depth Analysis.

### Q3: Are all components comprising GraphAdapter valid?

**A3: As Table 5 shows, removing any component of GraphAdapter results in performance drops.** Removing pre-training leads to an 0.91% drop, demonstrating that GraphAdapter's improvements indeed come from pre-training. Next, the most significant performance drop is when we simultaneously remove pre-training and graph structure in the fine-tuning stage (keeping only self-loops), which causes a 1.95% drop. This shows having the graph is crucial for GraphAdapter to work. Removing the task-related prompt leads to a 0.98% drop, validating our design of aligning pre-training tasks via prompts. Notably, removing the residual learning ("w/o Res Label" that is stated in section 4.2) leads to a 1.02% drop (more than removing pre-training), suggesting that training GNNs directly on all text may introduce excessive noise and hurt performance. Our Equation 7, which utilizes language model predictions to select words more semantically related to the graph, is reasonable.

However, although the ablation study validates the rationality of GraphAdapter's design and the efficacy of its components, these results hardly answer what exactly GraphAdapter pre-training is doing. Therefore, we further construct validation experiments about pre-training.

### Q4: What exactly does GraphAdapter's pre-training learn?

We conduct three comparative experiments to demonstrate what GraphAdapter pre-training is doing.

**(1) GNN can obtain stronger expressive power through pre-training.** We first observe the performance change of GNNs before and after pre-training, where we directly use the structural representations from the pre-trained GNN to fine-tune for downstream tasks. As Table 6 shows, the pre-trained GNN performs better on downstream tasks, improving by 0.78% on average. This demonstrates that GNNs are training their ability to model the graph structure during pre-training.

**Table 5: The performance of GraphAdapter when various components are removed. The evaluation metrics used for these tests align with those described above. The term 'w/o' indicates the removal of a specific component from the GraphAdapter.**

|  | Arxiv | Instagram | Reddit |
|---|---|---|---|
| w/o Pretraining | 0.7648 (0.0020) | 0.6392 (0.0086) | 0.6369 (0.0025) |
| w/o Graph structure | 0.7604 (0.0024) | 0.6346 (0.0074) | 0.6147 (0.0012) |
| w/o Res label | 0.7605 (0.0013) | 0.6408 (0.0130) | 0.6363 (0.0036) |
| w/o task-specific prompt | 0.7594 (0.0030) | 0.6364 (0.0073) | 0.6430 (0.0021) |
| GraphAdapter | 0.7707 (0.0015) | 0.6513 (0.0075) | 0.6461 (0.0019) |

**Table 6: The performance changes of the GNN block in GraphAdapter before and after pre-training. Here, "w/o Pretraining" signifies no pre-training, while "w Pretraining" indicates the opposite.**

|  | Arxiv | Instagram | Reddit |
|---|---|---|---|
| GNN w/o Pretraining | 0.7305 (0.0020) | 0.6181 (0.0112) | 0.6320 (0.0041) |
| GNN w Pretraining | **0.7335** (0.0024) | **0.6294** (0.0038) | **0.6410** (0.0027) |

**(2) Fusion block is learning how to fuse the knowledge from the language model and GNN during pre-training.** We further explore whether the fusion layer learned useful knowledge during training. We randomly initialize the parameters in a specific GraphAdapter's blocks after pre-trained. As Table 7 shows, initializing the parameters of the fusion layer leads to significant performance drops, decreasing by 1.03% on average across 3 datasets. Even on the Arxiv dataset, the performance is lower than full initialization. This result shows that the enhanced knowledge from GNN may need to be outputted through the matching fusion layer. To further verify this conjecture, we further reinitialized the parameters of GNN, and some performance decline can also be observed, decreasing by 0.82% on average. This is similar to the impact of reinitializing the fusion layer. The fusion layer alone does not contain much knowledge. Therefore, these results demonstrate that the fusion layer can learn how to fuse the knowledge from GNN and language models.

**(3) Graph structure is the basis of pre-training.** We further observe the changes in different base models before and after pre-training. In this comparative experiment, we keep all the structures of GraphAdapter, only replacing the GNN block with MLPs of equal parameter size. As Figure 3 shows, the MLP-based GraphAdapter shows no significant change before and after pre-training (average improvement of 0.19%), and even decreases in performance on Instagram and Reddit (drops of 0.05% and 0.62% respectively). While the GNN improves notably before and after pre-training (average improvement of 0.91%). This result suggests that GNN is a prerequisite for effective pre-training.

These three results demonstrate that GraphAdapter is indeed learning graph structures via pre-training. This validates that language-structure pre-training of GraphAdapter is reasonable and effective, and further supports the motivation of this paper.

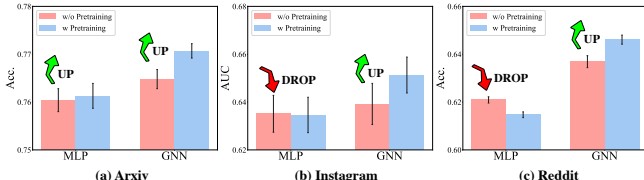

**Figure 3: The performance of GraphAdapter before and after pre-training, using MLP and GNN as the backbone architectures. The red represents performance without pre-training, while the blue represent performance after pre-training.**

**Table 7: The performance of GraphAdapter after randomly initializing some blocks. Here, "Re-init" represents re-initialization.**

|  | Arxiv | Instagram | Reddit |
|---|---|---|---|
| Re-init All | 0.7648 (0.0020) | 0.6392 (0.0086) | 0.6369 (0.0025) |
| Re-init GNN | 0.7680 (0.0022) | 0.6390 (0.0050) | 0.6364 (0.0026) |
| Re-init Fusion | 0.7562 (0.0011) | 0.6431 (0.0024) | 0.6378 (0.0022) |
| GraphAdapter | 0.7707 (0.0015) | 0.6513 (0.0075) | 0.6461 (0.0019) |

**Table 8: Running time of different methods on Arxiv using one Nvidia A800 80GB. Since different methods use different PLM, we also report the number of parameters for the PLM (decoded as "# para") and the number of trainable parameters ("# trainable").**

|  | GIANT | GLEM | GraphAdapter |
|---|---|---|---|
| PLM | BERT | DeBERTa-Large | Llama 2-13B |
| # para of PLM | 110M | 139M | 13B |
| # trainable in Pre | 110M | - | 3M |
| # trainable in Fine | 0.7M | 139M | 2M |
| Pre-process | - | - | 192 min |
| Pre-training | 341 min | - | 312 min |
| Fine-tuning | 1 min | 612 min | 1 min |
| Total time costs | 342 min | 612 min | 505 min |

## 5.4 Efficient

**Q5: How efficient is GraphAdapter?**

As shown in Table 8, we demonstrate the efficiency of GraphAdapter. As can be seen in Table 8, even when combined with a large model with 13B parameters, GraphAdapter has a speed comparable to PLM-based text-attributed graph modeling methods.

## 6 CONCLUSION

This paper proposes GraphAdapter to harness LLMs for TAGs without fine-tuning. A GNN adapter is trained to reduce LLM next-word errors on node texts. This adapts LLMs for graphs efficiently. Across node classification tasks, GraphAdapter improves accuracy by 5% over baselines. We validate with RoBERTa, GPT-2, and LLAMA 2, efficiently leveraging LLMs for interconnected text-graph data.

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

# A APPENDIX

## A.1 Dataset details

**Arxiv**. This paper uses the public partition, ground truth, and text information provided by OGB[15]. The few-shot train samples are sampled from the train set of public partition

**Instagram**. The original dataset for Instagram is provided by [18]. Since the original dataset did not contain graph information, we obtained users' follow lists, personal introductions, and tags for commercial users through Instagram's public API[3]. Therefore, the node text feature for Instagram is the user's personal introduction, and the edge represents the mutual relationship.

**Reddit**. Reddit is constructed on a public dataset [4] that collected replies and scores from Reddit users. The node text feature of this

---

[3]https://developers.facebook.com/docs/graph-api
[4]https://convokit.cornell.edu/documentation/subreddit.html

**Table 9: Detailed prompts on three datasets.**

| Dataset | Node feature | prompts |
|---------|--------------|---------|
| Arxiv | {abstract} | {This is a paper's abstract: [TEXT], this paper published on } |
| Instagram | {profile} | {This is a user's profile is: [TEXT], this user is} |
| Reddit | {content of last 3 posts } | {This is a user on Reddit, his last 3 posts are: [TEXT]. this user is} |

graph is the user's historical post content (limited to the last three posts per user), and the edge represents mutual replies between two users. We divided users into popular and normal categories based on their average score of history posts, with users whose average score is higher than the median considered popular and others considered normal.

## A.2 Prompts

According to the information on the downstream task and graph, this article has designed simple prompts for each dataset. As shown in Table 9. It should be noted that because PLMs are sensitive to prompts, different prompts may result in significant performance differences. However, how to find suitable prompts is not the focus of this paper, so no search for prompts is conducted.

## A.3 Implementation details

We independently pre-trained GraphAdapter on three datasets. The GNN used in the pre-training process was a 2-layer GraphSAGE, and the fusion layer used a 2-layer MLP. The pre-training was conducted for 50 rounds, and we used language model techniques such as silt activation function, layer-norm, and warm-up. The hidden side of GNN in GraphAdapter is set to 128, 64, and 128 on Arxiv, Instagram, and Reddit specifically

