# OpenReview forum: "GNNs as Adapters for LLMs on Text-Attributed Graphs"
_ACM.org/TheWebConf/2024/Conference — TheWebConf24_

### Official Review · Reviewer_jej8 · 2023-10-24

**Novelty:** 5
**Technical Quality:** 5

**Review:**

The paper introduces GraphAdapter, an innovative approach to harnessing the predictive power of large language models (LLMs) for Text-Attributed Graphs (TAGs). The paper aims to resolve the limitations of computational costs and representation power in jointly modeling text and graph structures. The authors propose an adapter GNN that works with pre-trained LLMs like RoBERTa, GPT-2, and Llama 2, showing computational efficiency and an average accuracy improvement of approximately 5% across multiple tasks and domains.

The following are three strength:
1. The paper provides an in-depth understanding of TAG challenges, laying a strong foundation for GraphAdapter's necessity and approach.

2. The paper tackles computational inefficiency by introducing a parameter-efficient GNN adapter, reducing trainable parameters significantly.

3. Comprehensive node classification experiments validate the model's effectiveness across multiple domains, showing a 5% accuracy improvement.

**Questions:**

1. Since the framework is efficient, I think there may be experiments on larger datasets, for example, Ogbn-Product
2. The prompt part seems also to play an important role in this framework. Nonetheless, there seems to be no detailed discussion about it.
3. The generative task in figure 2(a) is not clear for me

**Reviewer Confidence:**

3: The reviewer is confident but not certain that the evaluation is correct

**Scope:**

3: The work is somewhat relevant to the Web and to the track, and is of narrow interest to a sub-community

---

### Official Review · Reviewer_PixD · 2023-11-15

**Novelty:** 4
**Technical Quality:** 4

**Review:**

This work aims to address the challenges of high computational cost and insufficient model representation capacity in the joint learning of text and graph structures in TAG. The paper introduces its own solution, GraphAdapter, to tackle these issues. However, the entire article does not focus on addressing how the proposed method specifically resolves these problems. Instead, it is limited to introducing the method itself. The logical coherence in the Introduction section is lacking, as it fails to clearly articulate the problems it aims to solve. Additionally, the experimental section lacks comparative experiments with Large Language Models (LLM).

**Questions:**

1. in line 87-97 and line 99-107,
    > ...... However, when considering cascading GNN-LMs, existing
        techniques cannot be scaled up to billion-scale models like Llama
        2 [ 33 ] and GPT-3 [2]. Another pioneering research has ventured
        to fine-tune language models using unsupervised graph information .....

    >...... While cascading GNNs and LLMs prove infeasible for
        training, we draw inspiration from works on parameter-efficient
        tuning of LLMs to harness the power of large language models
        on TAGs .....

    The concatenation method incurs computational overhead, and self-supervised graph fine-tuning is introduced. You point out that graphs can assist language models in extracting node information, but it is not explicitly explained why this fine-tuning approach demonstrates that graphs help language models understand textual information. You've mentioned the computational cost of the concatenation method in your motivation. Then, you employ parameter fine-tuning, but you haven't clarified the difference between this and the self-supervised fine-tuning mentioned earlier. So, what is the innovation in your approach?

2. in line 117-132,
    > ...... we employ the GNN adapter only at the transformer’s
        last layer and implement residual learning for autoregressive next token prediction. Different from a traditional adapter, we perform        mean-pooling on the hidden representations from a GNN adapter  and LLMs, then optimize the adapter to improve the next-word prediction of the LLMs. .......

    Your main motivation is to address the computational cost associated with the concatenation method. However, the new approach you propose does not explicitly explain why using GNN in the last layer of the transformer reduces computational overhead. Additionally, while you suggest that graphs can help language models understand textual information, there is no structural innovation in your approach to enhance the effectiveness of graphs for language models. You haven't clarified the differences between the concatenation method and your approach in terms of how graphs assist language models, and why your method is superior. Therefore, what is your specific contribution in addressing these issues?

3. in line 317,
    > Text-Attributed Grap ......

    Your study focuses on text-attributed graphs, yet there is a lack of detailed introduction to this task. Instead, a substantial portion of the content is dedicated to explaining Pre-trained Language Models (PLM) and Graph Neural Networks (GNN), which may result in an illogical organization of the material. It's important to prioritize a comprehensive and clear explanation of the text-attributed graph task to ensure that readers understand the context and significance of your research.

4. in line 435-439,
    > ...... We separately calculated the prediction probabilities of the language model alone and the probabilities that mixed the graph structure and the previous predictions. The two robabilities are then averaged to obtain the final prediction result .......

    Given your assertion that language models struggle to predict graph-related word information, the decision to average the probabilities from the standalone language model and the graph-structured model raises several concerns:

            (1) The potentially poor performance of the standalone language model on graph-related words may adversely affect the overall effectiveness after averaging. Have you considered how the discrepancies in performance might be mitigated or addressed?
            (2) Averaging probabilities from language and graph models introduces additional computational steps, contradicting your initial motivation to address computational overhead. How does this align with the goal of reducing computational costs?
            (3) The rationale behind this specific approach, combining probabilities through averaging, is not clearly justified. Why choose this particular method of fusion, and how does it address the challenges posed by the difficulty of language models in predicting graph-related word information?
            Providing a more detailed explanation or rationale for this aspect of your methodology would help address these concerns and strengthen the coherence of your approach.

5. in line 209 and line 646,
    > ..... LLMs for Graph

    > table 2

    You rightly point out a potential gap in the evaluation of the proposed method. While the related work suggests the conversion of graphs to text for processing with Large Language Models (LLMs), there is a lack of comparative experiments in this aspect. The absence of experiments comparing the performance of the proposed method against approaches that directly utilize LLMs for graph processing makes it challenging to discern whether the improvement in model capability stems from the base model or from the fusion of graph and language model information.
    The absence of this comparison is also notable in the ablation experiment, where the experimental design falls short of demonstrating the clear advantages of the proposed model. Including experiments that specifically isolate and compare the contributions of the base model and the graph-language model fusion would enhance the robustness of your findings and better support your model's superiority claims.

6. in line 777,
    > Are all components comprising GraphAdapter valid?

    Table 5 indicates that the decrease in results is primarily attributed to pre-training. Removing both pre-training and graph structure, comparing the results with models that only exclude pre-training shows that the model's inference ability here is still predominantly derived from the base language model. This observation suggests that pre-training has a more significant impact on the overall performance than the exclusion of graph structures.

7. in line 866,
    > ...... Graph structure is the basis of pre-training .....

    You make a valid point regarding the extensive analysis of pre-training versus no pre-training models. The analysis focuses on the idea that pre-training enables the Graph Neural Network (GNN) to learn structural information from the graph. However, it's crucial to connect this back to the initial proposition in the introduction, where you suggested that in the Text-Attributed Graph (TAG) task, graphs can complement node text attributes through structural proximity.

    To strengthen your argument and provide a more comprehensive analysis, consider conducting sample analyses that highlight instances where the graph indeed supplements the text attributes of nodes. By examining specific examples, you can elucidate the cases where the graph contributes valuable structural information, thereby addressing the question of which samples benefit from the inclusion of graph structures. This, in turn, allows for a more nuanced understanding of the challenges the model faces and provides additional insights into the effectiveness of the proposed approach.

**Reviewer Confidence:**

3: The reviewer is confident but not certain that the evaluation is correct

**Scope:**

2: The connection to the Web is incidental, e.g., use of Web data or API

---

### Official Review · Reviewer_Lkj4 · 2023-11-23

**Novelty:** 4
**Technical Quality:** 4

**Review:**

This paper proposes a novel method, called GraphAdapter, which uses LLMs on graph structure data with parameter-efficient tuning. The method uses GNN as adapters for frozen LMs and pre-trains GNN to align with LMs.

Strength

It is interesting to use GNN as adapters for LLMs, so as to integrate LLMs with GNNs on Text-Attributed Graphs.

Weakness

1) The reported performance of baseline TAPE (GPT-3.5) in Table 3 on the Arxiv dataset (0.7672) differs from the original paper's result (0.7750 ± 0.0012), which seems unfair and the results are not competitive enough.

2) In the experiments, different prompts were set on different dataset. And the robustness of the prompt is lacking. Can another similar prompt achieve a comparable performance in GraphAdapter?

3) The paper is not well written and somewhat unclear.

**Questions:**

See the weakness.

**Reviewer Confidence:**

3: The reviewer is confident but not certain that the evaluation is correct

**Scope:**

4: The work is relevant to the Web and to the track, and is of broad interest to the community

---

### Official Review · Reviewer_vwz9 · 2023-11-29

**Novelty:** 4
**Technical Quality:** 5

**Review:**

The submitted work explores the use of LLM on Text-Attributed Graphs. The authors propose a GNN-based parameter-efficient tuning method for LLMs. They also propose a residual learning procedure to pre-train the GNN adapter with LLMs.

Pros
1. The idea of using the LLMs in structural and textual data is meaningful.
2. The proposed GNN Adapter is lightweight and convenient.
2. The paper is well-written and the experiments are sufficient.

Cons
1. Although the significant improvements in performance are shown in Table 2, the experimental results shown in Table 4 also show the limited performance on the dataset Arxiv and Reddit compared with the baseline GIANT when using the same LM.
2. As a parameter-efficient tuning method, the authors do not present the experimental comparison with popular parameter-efficient tuning, such as LoRA.

**Questions:**

1. What's the performance comparison of your method and baselines GIANT and GLEM with the same LLMs, such as (GPT2, and Llmam 2)?
2. What's the performance improvement of your GraphAdapter compared with the popular parameter-efficient tuning, such as LoRA?
3. What are the running time and GPU cost of your methods with different PLMs?
4. What's the effect of different GNNs with the same LM, such as graph attention networks?

**Ethics Review Description:**

Nan

**Reviewer Confidence:**

3: The reviewer is confident but not certain that the evaluation is correct

**Scope:**

2: The connection to the Web is incidental, e.g., use of Web data or API

---

### Decision · Program_Chairs · 2024-01-22

**Decision:**

Accept

**Comment:**

All the concerns of reviewers are addressed during the rebuttal, and all reviews are positive. I recommend a weak acceptance for this paper.